# An Attention Enhanced Spatial–Temporal Graph Convolutional LSTM Network for Action Recognition in Karate

**Jianping Guo [1], Hong Liu [2,*], Xi Li [2], Dahong Xu [2] and Yihan Zhang [1]**

1. College of Physical Education, Hunan Normal University, Changsha 410081, China; gjp2009@hunnu.edu.cn (J.G.); zyh@hunnu.edu.cn (Y.Z.)
2. College of Information Science and Engineering, Hunan Normal University, Changsha 410081, China; lixi@hunnu.edu.cn (X.L.); xudahong@hunnu.edu.cn (D.X.)
* Correspondence: liuhong@hunnu.edu.cn; Tel.: +86-139-0847-0936

**Abstract:** With the increasing popularity of artificial intelligence applications, artificial intelligence technology has begun to be applied in competitive sports. These applications have promoted the improvement of athletes' competitive ability, as well as the fitness of the masses. Human action recognition technology, based on deep learning, has gradually been applied to the analysis of the technical actions of competitive sports athletes, as well as the analysis of tactics. In this paper, a new graph convolution model is proposed. Delaunay's partitioning algorithm was used to construct a new spatiotemporal topology which can effectively obtain the structural information and spatiotemporal features of athletes' technical actions. At the same time, the attention mechanism was integrated into the model, and different weight coefficients were assigned to the joints, which significantly improved the accuracy of technical action recognition. First, a comparison between the current state-of-the-art methods was undertaken using the general datasets of Kinect and NTU-RGB + D. The performance of the new algorithm model was slightly improved in comparison to the general dataset. Then, the performance of our algorithm was compared with spatial temporal graph convolutional networks (ST-GCN) for the karate technique action dataset. We found that the accuracy of our algorithm was significantly improved.

**Keywords:** action recognition; Delaunay; karate; technical and tactical analysis; attention mechanism

## 1. Introduction

Artificial intelligence has been evolving for more than 60 years, driven by new theories and technologies such as mobile internet, big data, supercomputing, sensor networks, brain science, and the increasing demands of economic and social development. Artificial intelligence has become a new focus of international competition and a strategic technology for the future. In July 2017, the State Council of China issued a development plan for the new generation of artificial intelligence, putting forward the development goal that by 2030, the theory, technology, and applications of artificial intelligence in China will be world leading, making China the world's major artificial intelligence innovation center. Sports are related to national prosperity and national rejuvenation. Strong sporting performances can make a country strong, and successful national games can make a country prosperous. In line with the artificial intelligence national development goal, making full use of artificial intelligence technology in the development of sports, especially competitive sports, will be important. Based on research results from sports science and sports training, we should proceed with constructing sports big data, studying the rules of human intelligence activities in sports, constructing intelligent systems, and providing new high-tech means to improve athletes' competitive ability and promote public fitness.

Karate group hand sports, such as Taekwondo, Sanda, and various other sports, are types of fighting and confrontation events. Globally, and especially in Japan and Europe, karate has reached a high level after long-term development. On 3 August 2016, the

International Olympic Committee announced that karate would be an official event of the Tokyo Olympics. Our karate team will increasingly face global competition. Therefore, through tracking analysis and research into the skills and tactics of outstanding karate athletes at home and abroad, discovering the characteristics of the application of skills and tactics in karate group competitions, and exploring the winning rules of karate competitions, we can provide scientific guidance for athletes' in training and at competitions.

Technical and tactical analysis is very important in sports competitions. Traditional technical and tactical analysis, through watching training and competition videos, analyzes the characteristics of the athletes' technical movements, and their movement habits during the competition. This information is used to improve the efficiency and effectiveness of the athletes' training. In a game, this information can be used to formulate corresponding tactics according to the opponent's technical characteristics and habits. However, traditional technical and tactical analysis methods are undertaken by manually watching a large number of training and competition videos. High labor costs, serious data loss, long delays, and low accuracy restrict the efficiency of this type of technical and tactical intelligence analysis.

In recent years, some researchers have been trying to apply video-based action analysis technology to the analysis of competitive sport tactics and techniques, in order to improve the efficiency and accuracy of analysis. In this paper, our previously designed and improved ST-GCN algorithm [1] was applied to technical and tactical analysis and achieved good results. However, in our previously research work, it has been found that there are serious defects in applying existing techniques to karate technique and tactics analysis. First, the similarity between karate techniques is higher than that of other applications, and the ability of existing techniques to recognize such high similarity is severely reduced. Second, existing technology does not make full use of the prior information about the uniqueness of behaviors in the corresponding professional domain. Thirdly, through continuous research and experiments, it has been found that in the skeleton data of all kinds of technical movements of karate athletes, the importance or roles of key joint points of the human body are different for different types of movements. In view of this, on the basis of previous work, in this paper, the attention mechanism, long short-term memory (LSTM) [2], and Delaunay were combined to design a new graph convolution model. The model takes the information of athletes' joint points as input data, and different weight coefficients are assigned to the key joints. At the same time, it can obtain more abundant information related to the movement structure and spatiotemporal characteristics, which can effectively improve the accuracy of the model. On this basis, the automatic intelligent analysis of athletes' movement frequency statistics and trajectory tracking in technical and tactical analysis could be carried out. Artificial intelligence technology was integrated into traditional methods to achieve efficient technical and tactical analysis, so as to solve the problems of traditional technical and tactical analysis.

The main contributions of this work are summarized as follows:

(1) Using the Delaunay algorithm, a new composition algorithm based on a spatiotemporal sequence was designed. Based on the advantages of the Delaunay algorithm, we can obtain the spatiotemporal structural information of actions more effectively, and enhance the feature extraction ability of the attention-enhanced graph convolutional LSTM Networks (AGC-LSTM) [2].

(2) In order to truly apply the algorithm to the practical application of karate technology and tactics analysis, we constructed a small and medium dataset according to the requirements of the algorithm, which could be used for research on the karate technology and tactics analysis system based on action recognition.

## 2. Related Work

### 2.1. Karate Technical and Tactical Analysis

Analyzing and researching the techniques and tactics in sports training and competitions is an important part of scientific sports research. In recent years, experts and scholars have increasingly studied the techniques and tactics of karate competitions and training.

Zhou Yongsheng et al. [3] studied and summarized the evolution of karate competitive methods, and concluded that the evolution of karate competitive methods reflects the trend of modern competitive sports. In [4], it was proposed that karate has become an integral part of the life of college students in our country. The University of Shanghai for Science and Technology and Huzhou Normal University used questionnaire surveys and literature methods to study the cognition of college students after karate training, finding this is conducive to better promotion of karate projects. Scholars have also used comparative analysis research methods to analyze karate exercises. For example, Zhao Taojie and others [5] have used methods such as comparison and logical analysis for the determination of the anaerobic energy supply capacity in Taekwondo, Sanda, karate, and boxing. Starting from the importance of karate anaerobic endurance training, they expounded the effect of hand-shaping training in karate on the body's anaerobic glycolysis energy supply capacity. In [6], the 2017 National Competitive Karate Championship was used as the research object. Using the literature data method and mathematical statistics, along with other research methods, the authors concluded that the athletes in this competition have a higher level of skill than in previous competitions. Ren Jie [7], in order to study the regularity of sports injuries in karate, took karate athletes in Henan Province as a sample and obtained the injury rate, injury location, injury time, and injury causes among karate athletes. In [8], through the use of logic analysis, mathematical statistics, and expert interview methods, taking the 2016 National Karate Championship men's group hand athletes competition video as the research object, the common techniques used by national men's hand players were discovered, and the rules and characteristics of tactical use were summarized. Xie Jinwen et al. [9] used a literature review and other methods to assess the experience of Japanese karate entering the Olympics, and summarized the model and system of Japanese karate preparation for the Olympics, analyzed the athletes' technical performances, and provided basis and guidance for Chinese karate preparations for the Olympics. Feng Yajie et al. [10] used the literature review method and other research methods to study the 2018 National Karate Championship Finals, analyzing the current domestic competition pattern of karate events, and putting forward theoretical suggestions to support China's karate events at the 2020 Tokyo Olympics.

### 2.2. Action Recognition-Based Graph Convolutional Network

With the continuous development of camera technology, the training data for the action classification model was more accurate in processing the data. Compared with conventional video data, the human skeleton information in the video had less background interference. With the emergence of skeletal data, many scholars have tried to apply Graph Convolutional Networks (GCNs) to action classification based on human bones. GCNs and Convolutional Neural Networks (CNNs) have the same properties; they are both feature extractors, although CNN processing is a regular grid structure, and GCN can take the topological map of the human skeleton as input. Since 2018, GCN-based research has made good progress. In [11], a new dynamic skeleton model was proposed, namely the Spatial-Temporal Graph Convolutional Network (ST-GCN), which can automatically learn spatial and temporal patterns from skeleton data. The authors of [12] proposed a new two-stream Adaptive Graph Convolutional Network (2s-AGCN), which is used for skeleton-based action classification. In [13] the Actional-Structural Graph Convolution Network (AS-GCN) was proposed, which introduced an encoding–decoding structure to obtain the potential dependence of specific actions, or action links, using existing information to characterize high-order dependencies, or structural links, and learning the spatial and temporal characteristics of actions by superimposing action-structure graph convolution and time convolution. Si et al. [2] proposed a new attention-enhanced graph convolutional LSTM network (AGC-LSTM), which can not only capture the salient features of space and time, but also explore the co-occurrence relationship in the space–time domain. The current method fails to make full use of the topology structure of the skeleton graph, and only sets and fixes the skeleton node or adjacency matrix of the input sample manually,

which fails to solve this problem. In [14], an end-to-end architecture composed of the joints relation inference network (JRIN) and the skeleton graph convolutional network (SGCN) was designed. The JRIN can globally aggregate the spatiotemporal features of each joint, and then infer the optimal relationship of each joint, and the SGCN uses the optimal matrix for action recognition.

### 3. Model Architecture

For sequence modeling, many studies have proven that LSTM, as a variant of RNN, has an amazing ability to simulate long-term time dependence. Various LSTM-based models are used to learn the temporal dynamics of skeleton sequences. However, due to the full join operator in LSTM, the limitation of spatial correlation in skeleton-based action recognition is ignored. Compared with LSTM, AGC-LSTM [2] can not only capture the discriminant characteristics of spatial configuration and temporal dynamics, but also explore the co-occurrence relationship between the spatial domain and temporal domain.

The graph convolution model is widely used in sequence data based on skeleton nodes, and the construction of the graph model is the key to the graph convolution algorithm. Existing graph convolution models, such as AGC-LSTM, have problems such as a single graph structure, weak correlation between nodes, and insufficient differentiation of different actions.

In this paper, based on AGC-LSTM, the Delaunay triangulation method was used to establish the graphic model, so as to obtain more abundant feature information, strengthen the ability of the AGC-LSTM model to capture spatiotemporal feature information, and effectively improve the accuracy of the action recognition algorithm. The new algorithm proposed in this paper is called the attention enhanced spatial–temporal graph convolutional LSTM (ASTGC-LSTM). The details of the algorithm are given in detail below.

### 3.1. Attention Enhanced Spatial–Temporal Graph Convolutional LSTM

The ASTGC-LSTM also uses three gates: the input gate $i_t$, forgetting gate $f_t$, and output gate $o_t$. The input $X_t$, hidden state $H_t$, and cell memory $C_t$ are graph structure data, and the graph structure is generated by the Delaunay algorithm. Because of the graph convolution operator in the ASTGC-LSTM, cell memory $C_t$ and hidden state $H_t$ can obtain temporal dynamics, and contain spatial structure information. Figure 1 describes the structure of one ASTGC-LSTM layer. Figure 2 describes the structure of the ASTGC-LSTM unit. Equation (1) describes the function of ASTGC-LSTM unit.

$$
\begin{aligned}
i_t &= \sigma(W_{xi} * gX_t + W_{hi} * gH_{t-1} + b_i) \\
f_t &= \sigma\left(W_{xf} * gX_t + W_{hf} * gH_{t-1} + b_f\right) \\
o_t &= \sigma(W_{xo} * gX_t + W_{ho} * gH_{t-1} + b_o) \\
u_t &= \tanh(W_{xc} * gX_t + W_{hc} * gH_{t-1} + b_c) \\
C_t &= f_t \odot C_{t-1} + i_t \odot u_t \\
\widehat{H}_t &= o_t \odot \tanh(C_t) \\
H_t &= f_{att}\left(\widehat{H}_t\right) + \widehat{H}_t
\end{aligned}
\tag{1}
$$

where $* g$ denotes the graph convolution operator and $\odot$ denotes the Hadamard product. $\sigma(\cdot)$ is the sigmoid activation function. $u_t$ is the modulated input. $\widehat{H}_t$ is an intermediate hidden state. $W_{xi} * gX_t$ denotes a graph convolution of $X_t$ with $W_{xi}$. $f_{att}(\cdot)$ is an attention network that can select the discriminative information of key nodes. The $H_t$ as the output aims to strengthen the information of key nodes, not weaken the information of non-focus nodes, so as to maintain the integrity of spatial information.

The attention network adaptively focuses on key nodes through a software attention mechanism that automatically measures the importance of the key nodes. The principle of the spatial attention network is shown in Figure 2. The intermediate hidden state of the ASTGC-LSTM contains rich spatial structure information and temporal dynamics, which is

beneficial to guiding the selection of key nodes, so as to obtain the non-uniform degree weight sequence and strengthen the importance of different nodes in different types of actions.

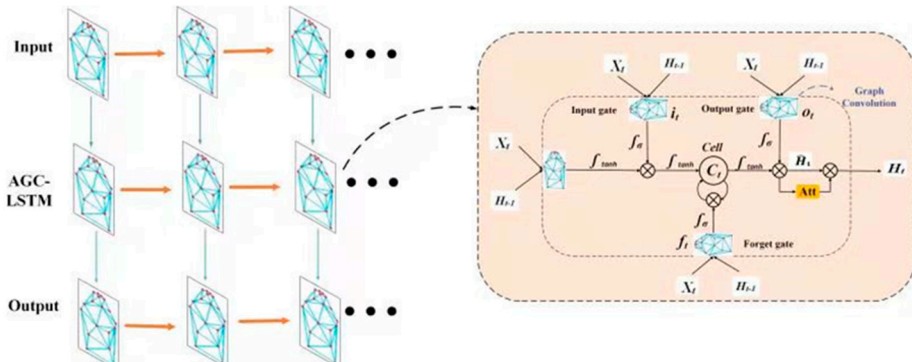

**Figure 1.** The structure of one ASTGC-LSTM layer.

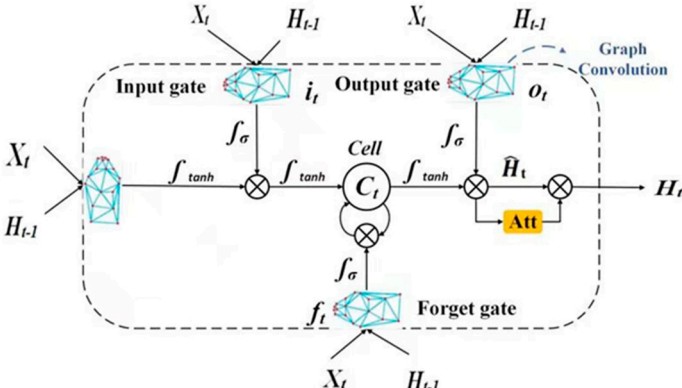

**Figure 2.** The structure of an ASTGC-LSTM unit. Compared with LSTM, the inner operator of the ASTGC-LSTM is graph convolution. In order to highlight more discriminative information, the attention mechanism is used to enhance the characteristics of key nodes.

The attentional network is used to meet critical points of attentional adaptation through a software attentional mechanism that automatically measures the importance of joints. The principle of the spatial attention network is shown in Figure 3. The intermediate hidden state of the ASTGC-LSTM contains rich spatial structure information and temporal dynamics, which is helpful to guiding the selection of key nodes, so as to obtain the weight sequence of the non-uniformity degree.

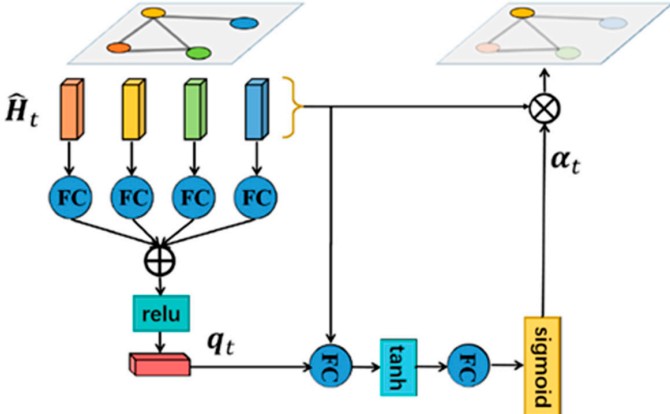

**Figure 3.** Illustration of the spatial attention mechanism principle.

### 3.2. Construction of the Spatial–Temporal Graph Based on the Skeleton

#### 3.2.1. Delaunay Algorithm

In the process of Delaunay triangulation, each point in the set of insertion points needs to undergo local optimization to judge the insertion points, so as to determine the local subdivision map.

The specific implementation process of the local optimization treatment is as follows:

(1) Two adjacent triangles form a quadrilateral;
(2) Make a circumferential circle for two triangles to check whether other points in the point set are in the circumferential circle;
(3) If a point in the point set is within the circumferential circle of any triangle, the diagonal is swapped to complete local optimization. This is shown in Figure 4.

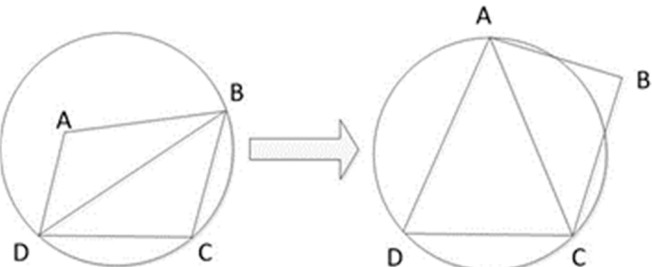

**Figure 4.** Process of local optimization.

Points A, B, C, and D in the figure represent four points in the point set, Triangle ABD and triangle BCD are two adjacent triangles, and the two triangles form a quadrilateral, ABCD. Point A is inside the circumferential circle of triangle BCD. The diagonal swapping rules are used to change the diagonal BD in the quadrilateral ABCD into diagonal AC, so that the circumferential circle does not contain any point in the point set.

#### 3.2.2. Spatial–Temporal Graph Model Based on Human Skeleton Nodes

The spatial–temporal Sequence of key points of the human skeleton obtained from the video sequence was used as a three-dimensional lattice to implement Delaunay three-dimensional triangulation. Three vertices of each triangle were taken as adjacency points to form a graph. Since adjacent triangles have common edges, the subgraphs constructed according to each triangle have common vertices, thus forming a stable graph structure. It can better describe the three-dimensional spatial structure information of a key point sequence and improve the identification degree of the action recognition model.

The resulting graph model is shown in Figure 5.

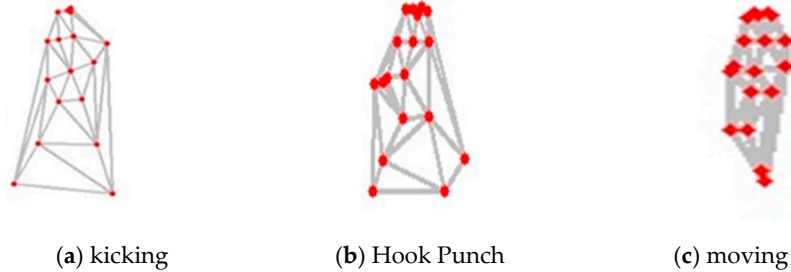

| (**a**) kicking | (**b**) Hook Punch | (**c**) moving |

**Figure 5.** Delaunay triangulation generated from the karate competition video.

### 3.3. ASTGC-LSTM Network

In this paper, an end-to-end attention enhancement spatial–temporal graph convolution LSTM network (ASTGC-LSTM) for skeleton-based human behavior recognition is described. In Figure 6, the overall pipeline for the novel model is shown. Some of the details of the framework are described in detail below.

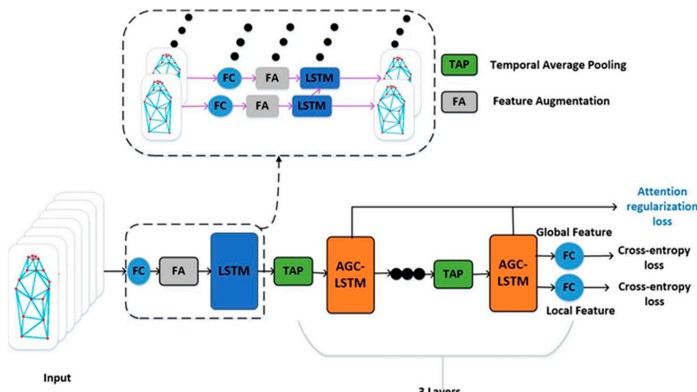

**Figure 6.** The architecture of the proposed ASTGC-LSTM network.

The human body can be divided into several parts according to its physical structure. The component-based ASTGC-LSTM network designed in this paper also has similar physical structure characteristics. First, the linear layer and shared LSTM layer were used to capture the feature information of each component. Then, the feature information of each component was input into the three layers of the ASTGC-LSTM as node representations in order to model the spatial–temporal features.

### 3.3.1. Spatial–Temporal Graph Model Based on Human Skeleton Joints

For the skeleton sequence, first a linear layer and LSTM layer were used to map the 3D coordinate of each joint onto a high-dimensional feature space. The first linear layer encodes the coordinates of the joint as a 256-dimensional vector, as the position features $P_t$, and $P_{ti}$ represent the position of joint i. Since it contains only position information, $P_{ti}$ is useful for learning spatial structure features in graph models. The frame difference feature $V_{ti}$ between two consecutive frames can facilitate the dynamic information for the ASTGC-LSTM. The series of two features can describe more abundant feature information, but the cascade of two features has the scale variance of the feature vector. Therefore, the LSTM layer was used to eliminate the scale variance between the two features. Equation (2) describes this principle.

$$
\begin{aligned}
E_{ti} &= f_{lstm}(concat(P_{ti}, V_{ti})) \\
&= f_{lstm}\left(concat\left(P_{ti}, \left(P_{ti} - P_{(t-1)i}\right)\right)\right)
\end{aligned}
\tag{2}
$$

where $E_{ti}$ is the augmented feature of joint *i* at time *t*.

### 3.3.2. Temporal Hierarchical Architecture

After the LSTM layer, the incremental features of the sequence $\{E_1, E_2, \cdots, E_T\}$ were input into the STGC-LSTM layer as node features. Three ASTGC-LSTM layers were stacked throughout the process to learn the spatial configuration and temporal dynamics. Inspired by the space pooling in CNN, the ASTGC-LSTM also adopts a temporal hierarchical architecture and average pooling to increase the temporal receptive field. Through the temporal hierarchical architecture, the temporal receptive field of each time input at the top ASTGC-LSTM layer becomes a short-term clip from a frame, which can be more sensitive to the perception of the temporal dynamics. Moreover, the performance of the algorithm is improved, and the computational complexity is greatly reduced.

### 3.3.3. Learning of the ASTGC-LSTM

Finally, the global feature $F_t^g$ and local features $F_t^l$ of each time step were converted into scores $o_t^g$ and $o_t^l$ of each class. According to Equation (3), the predicted probability of the $i^{th}$ class can be obtained.

$$\widehat{y}_{ti} = \frac{e^{o_{ti}}}{\sum_{j=1}^{C} e^{o_{tj}}}, i = 1, \dots, C \tag{3}$$

In the training process, considering that the hidden state of each time step on the top, the ASTGC-LSTM contains short-term dynamics, and the loss function in the form of Equation (4) was adopted into the training model.

$$
\begin{aligned}
L \quad &= -\sum_{t=1}^{T_3} \sum_{i=1}^{C} y_i \log \widehat{y}_{ti}^{g} - \sum_{t=1}^{T_3} \sum_{i=1}^{C} y_i \log \widehat{y}_{ti}^{l} \\
&+ \lambda \sum_{j=1}^{3} \sum_{n=1}^{N} \left(1 - \frac{\sum_{t=1}^{T_j} \alpha_{tnj}}{T_j}\right)^2 + \beta \sum_{j=1}^{3} \frac{1}{T_j} \sum_{t=1}^{T_j} \left(\sum_{n=1}^{N} \alpha_{tnj}\right)^2
\end{aligned}
\tag{4}
$$

where $y = (y_1, \dots, y_C)$ is the ground-truth label. $T_j$ denotes the number of time steps on the $j^{th}$ ASTGC-LSTM layer. The third term aims to pay equal attention to different joints. The last term is to limit the number of interested nodes. $\lambda$ and $\beta$ are weight decaying coefficients.

## 4. Intelligent Analysis and Experiment of Techniques and Tactics in Karate Videos

In this section, first, we compare our proposed ASTGC-LSTM with several state-of-the-art methods using the kinetics and NTU-RGB + D datasets.

Secondly, we compare our algorithm with ST-GCN on the karate technical action dataset. This is because the general karate training or competition videos were taken with ordinary cameras, and did not contain auxiliary data such as depth information.

### 4.1. Datasets

4.1.1. Kinetics Dataset

Kinetics-600 [15] is a large-scale, high-quality YouTube video URL dataset which contains all kinds of actions. The dataset consists of about 500,000 videos, covering 600 human actions, with at least 600 videos per action. Each video lasts about 10 s and is labeled with a class. These movements cover a wide range of human object interactions, such as playing musical instruments, and human to human interactions, such as handshakes and hugs. The skeleton data in the dataset was extracted by OpenPose [16]. All videos were adjusted to $340 \times 256$ resolution, and converted to 30 frames per second. As shown in Figure 7, the body 25 model in OpenPose detects 25 human joints.

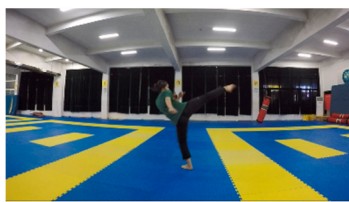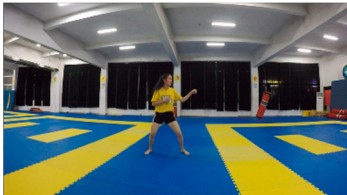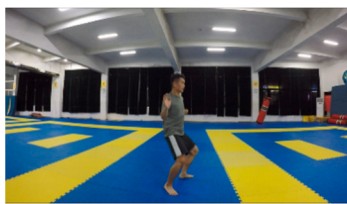

**Figure 7.** A frame of images corresponding to three types of actions in the dataset.

4.1.2. NTU-RGB + D Dataset

This dataset contains 60 different classes of human actions. This dataset was derived from a total of 56,880 action samples from 40 different subjects. Compared with the KINEC-TIC dataset, each sample in the NTU-RGB + D dataset not only contains RGB features, but also contains a series of depth maps, 3D skeleton data, and infrared videos, captured by three Microsoft Kinect V2 cameras at the same time. There are two evaluation protocols for this dataset: Cross Subject (CS) and Cross View (CV) [17]. In the NTU-RGB + D dataset, the actions of 20 subjects constituted the training set, and the actions of the other 20 subjects were used for testing.

### 4.1.3. Karate Technical Action Dataset

We filmed the training and competition videos of 10 athletes from the Hunan Karate Team to create a training and test dataset.

The dataset of karate technical action included the middle punch, upper punch, forward upper punch, back upper punch, back pounce, side rising kick, back kick, inside crescent kick, side kick, and nine types of technical movements. The actions of five athletes were collected from eight directions: east, south, west, north, southeast northwest, southwest, and northeast. There were 1847 video clips, including 696 kick clips, 625 fist clips, and 526 moving clips, each lasting about 10 s.

In this dataset, 1786 video clips were used as the training set, including 679 kicking clips, 608 Hook Punch clips, and 499 moving clips. Another 61 video clips were used as the test set, including 17 kicking clips, 17 Hook Punch clips, and 27 moving clips.

Figure 7 illustrates the corresponding video frames for kicking, hook punch, and moving. Figure 8 illustrates the video frames of three random perspectives in the test video.

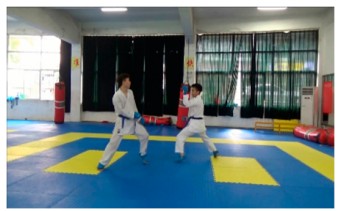 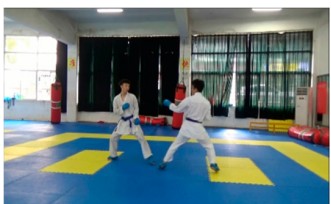 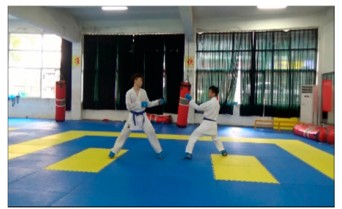

**Figure 8.** Three random perspectives from the test video.

### 4.2. Comparisons to State-of-the-Art Methods

#### 4.2.1. Experimental Analysis of the General Dataset

Using the kinetics and NTU-RGB + D datasets, the ASTGC-LSTM was compared with several other state-of-the-art methods [11–13,18]. Table 1 shows the top-1 and top-5 classification performances in the kinetics dataset. The experimental results showed that the ASTGC-LSTM is superior to other methods in the accuracy of the top-1 and top-5. Using the NTU-RGB + D dataset, the performance was slightly improved (Table 2).

**Table 1.** Performance comparison using kinetics with the current state-of-the-art methods.

| Methods | Top-1 Acc | Top-5 Acc |
|---|---|---|
| ST-GCN [11] | 30.7% | 52.8% |
| AS-GCN [13] | 34.8% | 56.6% |
| 2S-GCN [12] | 36.1% | 58.7% |
| Shift-GCN [18] | 37.5% | 60.2% |
| ASTGC-LSTM (ours) | 40.3% | 65.5% |

**Table 2.** Comparison with state-of-the-art methods, using the NTU-RGB + D dataset, for Cross-View (CV) And Cross-Subject (CS) evaluations of accuracy.

| Methods | CV | CS |
|---|---|---|
| ST-GCN [11] | 88.3% | 81.5% |
| AS-GCN [13] | 94.2% | 86.8% |
| HCN [19] | 91.1% | 86.5% |
| SR-TSL [20] | 92.4% | 84.8% |
| PB-GCN [21] | 93.2% | 87.5% |
| AGC-LSTM [2] | 95.0% | 89.2% |
| ASTGC-LSTM (ours) | 95.2% | 89.5% |

#### 4.2.2. Experimental Analysis of the Attention Mechanism of Karate Technical Actions

In the general dataset, many actions are highly distinguishable, such as standing up and combing hair. However, in the karate technique action dataset, there are a large number of highly similar actions. Most skeleton-based action recognition algorithms assign the same weight to the joints, resulting in a higher recognition error rate for these algorithms

in the karate technique action dataset. In fact, in different types of technical actions, each key joint has a different expression ability, so the joints that have a strong ability to express the action need to be given a higher weight.

In this paper, the attention mechanism was integrated into the model to solve this problem. Figure 9 shows the degree of attention of two actions with high similarity in the ASTGC-LSTM layer. The difference between a forehand punch and backhand punch is that one punches with the left hand and the other punches with the right hand. It is difficult to distinguish these two actions in a graph convolution of the topology. However, we can see in Figure 7 that the forehand punch and backhand punch have significant differences in the attention weight of the joints. The above results showed that the proposed ASTGC-LSTM is an effective skeleton-based action recognition method.

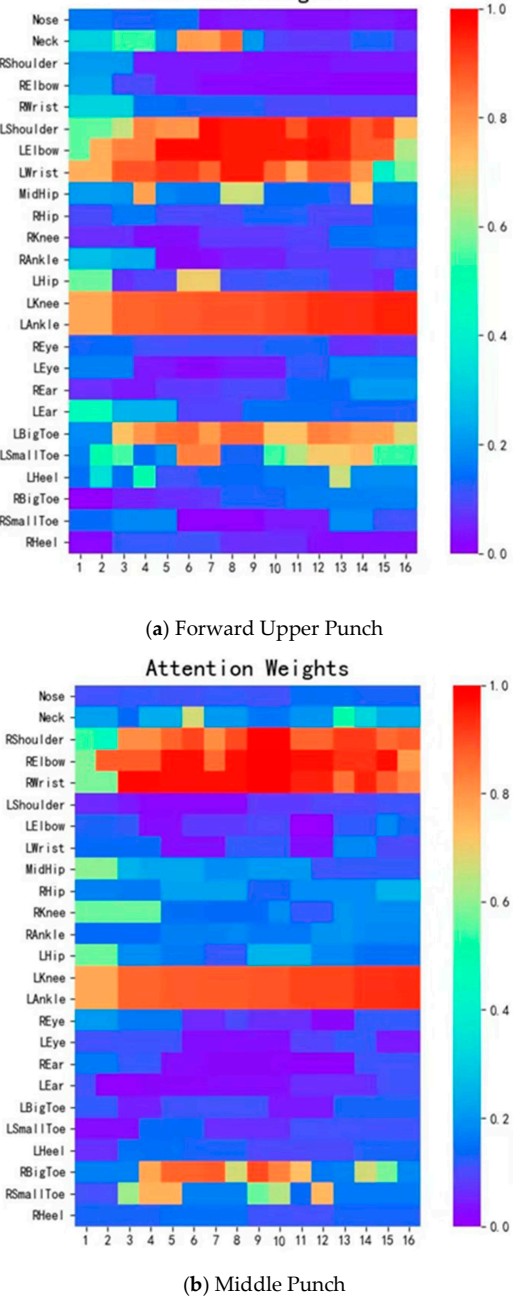

(**a**) Forward Upper Punch

(**b**) Middle Punch

**Figure 9.** Karate technique attention mechanism heat map. The *x*-axis is the frame of the video, and the *y*-axis is the action category.

### 4.3. Comparative Analysis with the Karate Technical Action Dataset

#### 4.3.1. Topology Comparison

Based on the basic framework of [11,13], the topology was constructed using the Delaunay triangulation method. The ST-GCN [11], as shown in Figure 10a, is a simple joint point connecting itself. The AS-GCN [13], as shown in Figure 10b, can associate and connect potentially relevant joints in a specific action. As shown in Figure 10c, this paper proposed a graph construction method to obtain the topology of the video frame through the Delaunay triangulation algorithm for different actions.

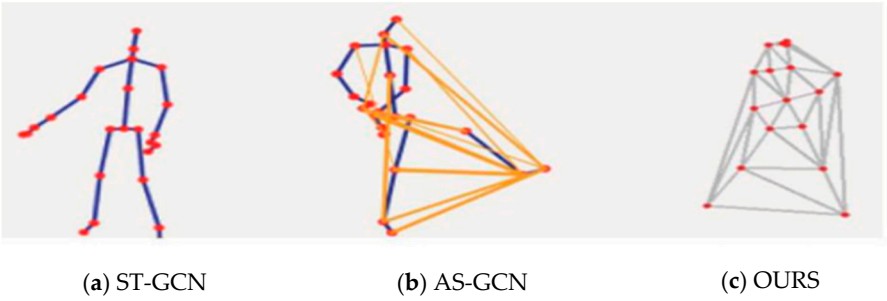

(**a**) ST-GCN      (**b**) AS-GCN      (**c**) OURS

**Figure 10.** Topology of the ST-GCN, AS-GCN, and our algorithm.

#### 4.3.2. Performance Comparison

Since the karate technique action video only contained RGB features, we used the ST-GCN and our algorithm for comparison with the karate technique action dataset. Figure 11 shows the comparison between the feature representation in the ST-GCN and our algorithm.

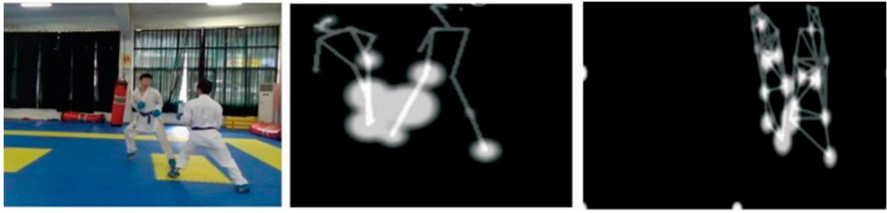

**Figure 11.** Comparison of feature representation between the ST-GCN and our algorithm.

White circles indicate the relevance of features. The larger the circle, the stronger the relevance of a joint point to a certain action; the more circles, the more related feature points extracted from a certain action.

As shown in Figure 11c, the constructed topology can obtain more related joints, so as to obtain more abundant action features.

As shown in Figure 11b,c, in the process of "kicking", the feature of the right arm in the ST-GCN was significantly weakened, while the feature responsiveness in this paper still persisted. Compared with the ST-GCN, there were more joint points in the kick action, which provided more abundant features for action.

Using the karate technical action dataset, due to the complexity of karate technical actions, the accuracy of the ST-GCN was much lower than the results using the kinetic dataset. In our algorithm, the attention mechanism is integrated into the topology, making the accuracy of this algorithm much better than that of the ST-GCN, as shown in Table 3.

**Table 3.** Performance comparison with the karate technical action dataset of the ST-GCN and our algorithm.

| Methods | Top-1 Acc | Top-5 Acc |
| --- | --- | --- |
| ST-GCN [11] | 25.2% | 40.9% |
| ASTGC-LSTM (ours) | 45.2% | 68.6% |

## 5. Conclusions

In this paper, Delaunay triangulation was used to construct the topological graph in the graph volume model, and we proposed an action recognition algorithm based on the new graph convolution model. At the same time, the algorithm was used to realize karate competition video technique and tactics analysis. Through a large amount of experimental analysis we found:

(1) The new method of constructing graphs had a larger information capacity, which was more conducive to the accuracy of action recognition.
(2) The new action recognition algorithm improved the intelligence and accuracy of the technical and tactical analysis.

**Author Contributions:** Conceptualization: J.G. and H.L.; Funding acquisition: J.G.; Investigation: X.L. and D.X.; Methodology: J.G. and H.L.; Project administration: D.X. and X.L.; Software: D.X. and X.L.; Visualization: X.L and Y.Z.; Writing—original draft: J.G. All authors have read and agreed to the published version of the manuscript.

**Funding:** This work is supported by the research on image restoration algorithm based on regularization method (No. 10JJ3060).

**Conflicts of Interest:** The authors declare no conflict of interest.

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
