# Peer review of "An Attention Enhanced Spatial–Temporal Graph Convolutional LSTM Network for Action Recognition in Karate"

_applsci, doi:10.3390/app11188641_

Round 1

Reviewer 1 Report

In this paper, the authors tackled an interesting problem of action recognition using deep learning, specifically recurrent neural nets that benefit from long short term memory cells and attention mechanism. Overall, the manuscript is well-written and sound, but addressing the following issues would help improve its quality further:

  1. All acronyms should be defined at their first use (see e.g., ST-GCN in the abstract). Also, I suggest removing such acronyms from the abstract as they adversely affect the read.
  2. I encourage the authors to present the motivation behind this work in more detail in the introductory section, especially given that it originates from the previous approach already published in the literature. Specifically, it would be useful if the authors could highlight the shortcomings of existing techniques together with the open issues from the literature (i.e., please expand the discussion from lines 70-72) – it would help better highlight the contributions of this work.
  3. The quality of the figures is rather poor and should be improved.
  4. There are vague statements across the manuscript (also in the captions of the figures – see e.g., Figure 4). Also, please avoid having one sentence long paragraphs (e.g., lines 323).
  5. The multiplication operation should be indicated as \cdot rather than *, as * is commonly used for convolution. Also, when the authors refer to the spatial resolution, e.g., of a video, I suggest replacing * with \times.
  6. What is the approach for dividing the investigated datasets into training and test samples (see e.g., line 331)? Were these divisions kept unchanged for all investigated algorithms? This should be clearly discussed in the experimental part of the manuscript.
  7. The proposed algorithm would not be trivial to reimplement, hence the experiments are very difficult to reproduce (the same applies to the training/test datasplits – they should be made publicly available if they are not at the moment). To this end, I strongly encourage the authors to make their implementation publicly available, preferably with a minimal example of executing the code – it would allow other research groups to reproduce the experiments and to seamlessly compare the emerging approaches with the proposed technique over the very same data.
  8. Would it be beneficial to exploit the gated recurrent units instead of the LSTM cells in the proposed architecture, as the former are more compact? It would be useful to see a minimal experiment showing the impact of replacing LSTM cells with the GRU counterparts. Also, it would be useful if the authors could execute an ablation study in order to better understand the impact of specific architectural choices presented in the architecture on its overall abilities.
  9. Although the manuscript reads well in general, it would benefit from proofreading (e.g., there are open quotation marks, see e.g., line 37).

Author Response

1. The full name of the abbreviated term that appears for the first time has been supplemented in the full text.

2. Near lines 70-72, three shortcomings in the analysis of karate techniques and tactics of the existing technology have been added, which shows the pertinence and importance of the work of this paper.

3. The poor quality pictures in the original paper have been replaced with clearer pictures. (Figure 7 and Figure 8).

4. This paper has been handed over to the editing organization designated by the editing services (https://www.mdpi.com/authors/englishthe) for full text editing. This paper has been revised and certified.

5. The problem of using * to describe the image resolution in line 311 of the original paper has been modified.

6. In order to increase the accuracy of the algorithm evaluation and the generalization of the model, it is necessary to increase the training samples and test samples as much as possible.

7. The code has been uploaded to github. (https://github.com/gjp2009/ASTGC_LSTM).

8. We are currently doing a large number of ablation experiments and plan to make further improvements to the algorithm modules in the follow-up work.

9. The grammatical errors in the paper have been corrected by the agency designated by the editorial department.

Reviewer 2 Report

In this paper, authors use machine learning, specifically graph convolutional LSTM networks, to recognize Karate actions in order to better analyze the game.

The authors did a good job explaining their method and comparing it to the state of the art.

The main weaknesses of the paper are the following:

  • As a reader, I still don't understand why it would be that important to recognize Karate actions.
  • The authors were not able to achieve very satisfying accuracy results with their method, even though it might be better than the other work they compared to.
  • The authors did not try other machine learning techniques that could have been a better fit for their application.
  • The English in the paper needs to be revised a little as it has some grammatical errors.

Author Response

1. In the technical and tactical analysis of all competitive sports, it is necessary to analyze a large amount of information related to the technical movement of a specific sport, such as the frequency of use of the movement and so on. The basis of these analyses is to classify and identify the technical movements of specific athletes.

2. Compared with the existing general-purpose behavior recognition algorithms, the algorithm designed in this paper has been applied to the analysis of karate techniques and has significantly improved the recognition ability, but it has not yet reached the practical application effect.

3. This paper has been handed over to the editing organization designated by the editing services (https://www.mdpi.com/authors/englishthe) for full text editing. This paper has been revised and certified.

Round 2

Reviewer 1 Report

I am happy to see that the authors have addressed my concerns, and the manuscript is in a better shape now.